# Differences between SCORE, Framingham Risk Score, and Estimated Pulse Wave Velocity-Based Vascular Age Calculation Methods Based on Data from the Three Generations Health Program in Hungary

**DOI:** 10.3390/jcm13010205

**Published:** 2023-12-29

**Authors:** Helga Gyöngyösi, Gergő József Szőllősi, Orsolya Csenteri, Zoltán Jancsó, Csaba Móczár, Péter Torzsa, Péter Andréka, Péter Vajer, János Nemcsik

**Affiliations:** 1Department of Family Medicine, Semmelweis University, 1085 Budapest, Hungary; helgagyongyosi@gmail.com (H.G.); moczar.csaba@semmelweis.hu (C.M.); torzsa.peter@semmelweis.hu (P.T.); 2Gottsegen National Cardiovascular Center, 1096 Budapest, Hungary; szolgerjozs@gmail.com (G.J.S.); csenteri.orsolya@gmail.com (O.C.); jancso.zoltan.g3@gmail.com (Z.J.); peter.andreka@gokvi.hu (P.A.);; 3Coordination Center for Research in Social Sciences, Faculty of Economics and Business, University of Debrecen, 4032 Debrecen, Hungary

**Keywords:** vascular age, risk scores, estimated pulse wave velocity

## Abstract

Early vascular ageing contributes to cardiovascular (CV) morbidity and mortality. There are different possibilities to calculate vascular age including methods based on CV risk scores, but different methods might identify different subjects with early vascular ageing. We aimed to compare SCORE and Framingham Risk Score (FRS)-based vascular age calculation methods on subjects that were involved in a national screening program in Hungary. We also aimed to compare the distribution of subjects identified with early vascular ageing based on estimated pulse wave velocity (ePWV). The Three Generations for Health program focuses on the development of primary health care in Hungary. One of the key elements of the program is the identification of risk factors of CV diseases. Vascular ages based on the SCORE and FRS were calculated based on previous publications and were compared with chronological age and with each other in the total population and in patients with hypertension or diabetes. ePWV was calculated based on a method published previously. Supernormal, normal, and early vascular ageing were defined as <10%, 10–90%, and >90% ePWV values for the participants. In total, 99,231 subjects were involved in the study, and among them, 49,191 patients had hypertension (HT) and 15,921 patients had diabetes (DM). The chronological age of the total population was 54.0 (48.0–60.0) years, while the SCORE and FRS vascular ages were 59.0 (51.0–66.0) and 64.0 (51–80) years, respectively. In the HT patients, the chronological, SCORE, and FRS vascular ages were 57.0 (51.0–62.0), 63.0 (56.0–68.0), and 79.0 (64.0–80.0) years, respectively. In the DM patients, the chronological, SCORE, and FRS vascular ages were 58.0 (52.0–62.0), 63.0 (56.0–68.0), and 80.0 (76.0–80.0) years, respectively. Based on ePWV, the FRS identified patients with an elevated vascular age with high sensitivity (97.3%), while in the case of the SCORE, the sensitivity was much lower (13.3%). In conclusion, different vascular age calculation methods can provide different vascular age results in a population-based cohort. The importance of this finding for the implementation in CV preventive strategies requires further studies.

## 1. Introduction

Cardiovascular (CV) diseases are still the leading cause of global morbidity and mortality. Proper medication and lifestyle changes can lead to a reduction in their adverse impacts, in particular in the case of hypertension [1]. The cornerstone of primary prevention is to identify high-risk patients. The calculation of CV risk and its communication by the physician may make patients aware of the possible consequences, and they can become more motivated about healthy lifestyles and the long-term use of proper medication. Demonstration of the absolute CV risk is not always convincing as the numeric value that represents this risk in percentage can be relatively low with the risk of misinterpretation. The concept of vascular age was created to demonstrate if the patient’s vasculature is older than their chronological age, which might be more convincing for long-term adherence [2].

The Framingham Risk Score (FRS), which is a widely used method for stratifying CV risk, was introduced in 2008. It was derived from the data of 8491 participants in the Framingham study [2]. Over a 12-year follow-up period, 1174 participants experienced their first CV event. FRS offers sex-specific estimates of an individual’s 10-year risk of developing a fatal or non-fatal CV event. The paper that introduced FRS also presented the concept of vascular age calculation. This approximation is based on first the calculation of the FRS for an individual and then determining the age of a person with the same predicted risk but with all other risk factors falling within the normal ranges [2].

The Systematic Coronary Risk Evaluation (SCORE) is also a commonly used tool for estimating the 10-year risk of fatal CV events. This estimation is based on a combined database from 12 European cohort studies, primarily conducted in the general population. The database includes 205,178 subjects, corresponding to 2.7 million person–years of follow-up, during which 7934 CV deaths occurred, including 5652 from coronary heart disease [3]. Additionally, a method for calculating vascular age based on the SCORE was also published [4]. The definition of vascular age in the SCORE shows similarities with the vascular age concept in the FRS. In both cases, it represents the age of a person with the same CV risk but with all risk factors falling within the normal ranges. Essentially, this represents a CV risk solely attributed to age and gender [4].

Aortic stiffness, as measured by carotid–femoral pulse wave velocity, is a standalone predictor of future events in individuals with hypertension In the Systolic Blood Pressure Intervention Trial (SPRINT) population, estimated pulse wave velocity (ePWV) was a reliable predictor of the primary outcome and all-cause death, independently of the FRS [5].

There are previous studies which have investigated the differences between the calculated vascular ages evaluated with CV risk-based methods. In one of our previous papers published in the topic and evaluating 172 participants, significant differences were observed in the vascular age values calculated with the FRS or SCORE, as well as in the proportion of individuals with impaired vascular age when assessed with the measured PWV, FRS, or SCORE [6].

The aim of our study was to compare SCORE and FRS-based vascular age calculation methods and their relation to early vascular ageing based on ePWV on a huge population-based sample in Hungary.

## 2. Materials and Methods

### 2.1. Three Generations for Health Program

The Three Generations for Health program is organized by the National Directorate General for Hospitals in partnership with the Gottsegen György National Cardiovascular Center of Hungary. The main objectives of the program focus on reducing mortality rates related to coronary heart disease and CV ailments, exhibiting a seamless alignment with Health Sector Strategy of Hungary’s goals [7]. The main goals of the Three Generations for Health program are the evolution of primary health care and collaboration between the participants involved in general practitioners’ services through cooperation with the health promotion offices and also local governments [7]. Alongside evaluating the population’s CV risk, another goal of this program is to introduce healthy lifestyle practices and tools of primary prevention. The program involves 806 general practitioner practices across the country; the initiative targets three generations of participants (0–18 years, 40–65 years, and 65+ years) and aims to reduce mortality from coronary heart disease and CV diseases in alignment with Hungary’s Health Sector Strategy. Therefore, this initiative embodies a comprehensive strategy for assessing cardiovascular risk, combining medical methodologies, technological advancements, and strategic alignment with national health goals.

The study was performed in respect of the guidelines of the Declaration of Helsinki with the respect of the Regulation 2016/679—Protection of natural persons regarding the processing of human data. All participants enrolled in the Three Generations for Health program provided their informed consent. According to the Medical Research Council, no approval from an ethics committee was needed as all procedures were performed in compliance with the applicable standards and regulations, paying full regard to the decision made by the government.

### 2.2. Calculation of Vascular Age Using the Framingham Risk Score 

The calculation of vascular age using the FRS was carried out following the methods of D’Agostino et al., which details the process of FRS calculation [2]. The calculation provides sex-specific results and considers age, total cholesterol, high density lipoprotein cholesterol (HDL), brachial systolic blood pressure, ongoing smoking, and the presence of diabetes or treated hypertension. The original publication also provides an estimation of vascular age, first with FRS calculation and next with the calculation as the age of a subject with the same risk but with all risk factor levels in the normal ranges [2]. In the FRS vascular age calculation, the highest value is designated as ‘80+’ but in our calculations, we considered the age of 80 years for these participants.

### 2.3. Calculation of Vascular Age Using the Systematic Coronary Risk Evaluation (SCORE) Risk Score

The SCORE risk score calculation considers age, sex, smoking status, brachial systolic blood pressure, and total cholesterol, and it is different in low and high CV risk European countries [3]. The procedure for estimating vascular age using the SCORE framework shows similarities with the FRS-based method, as the vascular age of the subject is equal to the age of somebody with the same CV risk but without any risk factors, meaning a risk only due to gender and age. The calculated vascular ages with the two SCORE charts (designed for high- and low-risk countries) were in high agreement, suggesting the widespread applicability of this concept [4]. 

### 2.4. Calculation of ePWV

The equation of the ePWV was described in the study of Greve et al. [8] and was derived by the Reference Values for Arterial Stiffness’ Collaboration [9]. Age and MBP were used to evaluate ePWV following the formula:ePWV = 9.587 − 0.402 × age + 4.560 × 10^−3^ × age^2^ − 2.621 × 10^−5^ × age^2^ × MBP + 3.176 × 10^−3^ × age × MBP − 1.832 × 10^−2^ × MBP.

The mean BP was calculated as diastolic BP (DBP) + 0.4 (SBP − DBP).

### 2.5. Supernormal, Normal, and Early Vascular Ageing 

Supernormal, normal, and early vascular ageing were defined as <10%, 10–90%, and >90% ePWV values for the participants, following the study of Bruno RM et al. [10].

### 2.6. Statistical Analysis

Categorical data are presented as frequencies and as medians and interquartile ranges for data measured on a continuous scale. Data analysis was performed using chi-squared tests for categorical variables. Data measured on a continuous scale were analyzed using Kruskal–Wallis tests due to the non-symmetric distribution of the data in all cases. The *p*-values from statistical analyses were considered significant if the *p*-values from the procedure were less than 0.05. Stata Statistical Software (version 13.0, Stata Corp, College Station, TX, USA) was used for the statistical analysis, and *p* <  0.05 was considered significant.

## 3. Results

During the data collection period between January 2019 and January 2022, 99,231 subjects were involved in the study, and among them, 49,191 patients had hypertension (HT) and 15,921 patients had diabetes (DM). Table 1 provides the demographic data and baseline laboratory parameters. 

The median chronological age in the whole cohort was 54.0 (48.0–60.0) years, the median vascular ages calculated with the SCORE and FRS were 59.0 (51.0–66.0) and 64.0 (51.0–80.0) years, respectively. In the HT patients, the chronological, SCORE, and FRS vascular ages were 57.0 (51.0–62.0), 63.0 (56.0–68.0), and 79.0 (64.0–80.0) years, respectively (*p* < 0.05). In the DM patients, the chronological, SCORE, and FRS vascular ages were 58.0 (52.0–62.0), 63.0 (56.0–68.0), and 80.0 (76.0–80.0) years, respectively (*p* < 0.05). In participants without HT or DM, the chronological, SCORE, and FRS vascular ages did not differ in clinically significant manner (51.0 (45.0–57.0), 54.0 (47.0–62.0), and 51.0 (45.0–64.0), respectively). Figure 1 demonstrates the chronological and vascular ages calculated with the SCORE and FRS in the whole group, in HT patients, in DM patients, and in participants without HT or DM.

Based on our previous publication [6], groups of subjects were created based on the arbitrary threshold of a 2-year difference between SCORE vascular age or FRS vascular age, compared with chronological age defining the following groups: -Age difference < −2 years: SCORE−, FRS−: supernormal vascular ageing;-Age difference between −2 and 2 years: SCORE normal, FRS normal: normal vascular ageing;-Age difference > 2 years: SCORE+, FRS+: early vascular ageing.

Based on this definition, 17,210 (17.3%) and 10,608 (10.7%) of the patients fell into the normal vascular ageing category according to the SCORE and FRS, respectively. A total of 57,433 (57.9%) subjects were SCORE− and 24,588 (24.5%) subjects were SCORE+. Based on FRS vascular age calculation, 18,659 (18.8%) patients were FRS− and 69,964 (70.5%) patients were FRS+.

The ePWV in the total population, HT patients, and DM patients was 9.0 (8.1–10.0) m/s, 9.6 (8.7–10.3) m/s, and 9.7 (8.8–10.4) m/s, respectively. Table 2. describes the characteristic of the supernormal, normal, and early vascular ageing patients based on ePWV. In all ePWV-based vascular ageing categories, FRS vascular age was higher compared with SCORE vascular age (*p* < 0.05). Among the patients with early vascular aging, 2718 (27.7%) had neither HT nor DM. 

Table 3 demonstrates the overlap between participants identified as having supernormal, normal, and early vascular ages with the three different methods. The FRS identified patients with EVA based on ePWV with higher sensitivity (97.3%) compared to the SCORE (13.3%); however, the ratio of the FRS+ patients was high in all vascular ageing categories. Differences between the SCORE and FRS-based vascular ageing categories were significant in all settings (*p* < 0.05).

Elevated vascular age was determined based on the SCORE and FRS in 24.8%, and 70.5%, respectively. However, only 3.9% of the FRS+ patients were SCORE+ as well, and in SCORE+ patients, the overlap with FRS+ was 11%. Approximately 13.7% of the FRS+ patients were found to have early vascular ageing based on ePWV, and only 5.3% of the SCORE+ patients were confirmed by ePWV. Approximately 97.3% of the ePWV+ patients were FRS+; on the other hand, only 13.3% of the ePWV+ subjects were SCORE+. With all three methods, only 1.1% of the subjects were found to be older than their chronological age. Figure 2 demonstrates the overlap between subjects identified with early vascular ageing through the use of different methods.

## 4. Discussion

This is the first study, which confirmed our previous findings in a population-based cohort using SCORE and FRS-based vascular age calculation methods, to identify different patients with early vascular ageing. Marked differences were found between the vascular age values calculated with the FRS or SCORE in patients with diabetes and hypertension, suggesting that in these conditions, the FRS-based method could be more convincing in risk communication. Based on ePWV, subjects with early vascular ageing were identified with high sensitivity using the FRS-based method, while only a small proportion of the participants were identified by the SCORE. 

Our findings about the differences between FRS-based and SCORE-based vascular age calculation methods in patients with treated hypertension or diabetes are in line with previous findings. The FRS-based method resulted in a markedly higher vascular age in those treated for hypertension and diabetes in two of our previous studies, with 172 and 241 subjects involved [6,11], and in the study of Kozakova et al. including 528 participants [12]. In that study, the FRS vascular age was higher compared to the SCORE vascular age in the early vascular ageing group which was based on the common carotid artery distensibility coefficient [12]. The present, population-based study also confirms these findings, suggesting that in these conditions, the FRS-based method might be more convincing in CV risk communication than the SCORE-based method. In subjects free of hypertension or diabetes, however, there were significant differences between the groups, but the clinical significance is moderate, so the two methods might be used interchangeably. 

In our present, population-based study, only minor overlap was found between patients with early vascular ageing evaluated with the different methods. This finding is in line with two of our previous studies. Vecsey-Nagy et al. found that 83.4% and 93.8% of the population had an elevated vascular age based on the FRS and SCORE, respectively, while only 42.3% had an elevated vascular age based on CACS. Approximately 38.2% of the patients were found to be older than their chronological age with all three methods [11]. In the other previous study of our lab, 78.5% and 32% of the population were found to be FRS+ and SCORE+ and 40.1% were PWV+. Approximately 9.3% of the subjects’ vascular ages were found to be higher than their chronological age with all three methods [6]. These findings confirm marked differences between the identified patients with early vascular ageing and warrants further studies and consensus to overcome these methodological differences.

Vascular age calculation methods can also be divided into functional and morphological measurement-based categories, besides risk score-based ones. In line with our findings, recent publications have discovered marked differences between functional and morphological methods as well. Both in the study of Yurdadogan T et al. and Sigl M et al., in a high proportion of subjects, significant differences were found between the PWV-based (functional) and carotid artery intima–media thickness-based (morphological) methods in the identification of early vascular ageing [13,14]. Besides these emerging problems, the hypothesis that vascular ageing communication is superior to communication of the absolute CV risk in percentage has not been proven yet. Moreover, in low-risk patients, an internet-based survey with a short-term follow-up (2 weeks) failed to prove the superior effectiveness of vascular age communication on the intention to change one’s lifestyle [15]. However, a recent review paper nicely summarizes the clinical potential of the vascular ageing concept and the added value of vascular ageing biomarkers to established biomarkers in the prediction of CV outcomes [16]. 

In contrast with the clinical use of risk scores, namely SCORE2 and SCORE2-OP, which are strongly recommended (class of recommendation: I, level of evidence: B) both in the recent CV prevention and hypertension guidelines [17,18], the use of vascular age as an alternative risk communication strategy is not involved in guidelines. Maybe by fulfilling the gaps in evidence and confirming the effectiveness of the clinical utility of different methods, the vascular age concept could have a place in future guidelines. 

The marked differences between the calculated vascular ages can rely on the methodological characteristics. The SCORE risk calculator includes a lower number of risk factors compared with the FRS and considers the risk of 10-year CV mortality, which might underestimate the absolute risk of a young patient with several CV risk factors [4]. Otherwise, the FRS integrates treated risk factors as well, like hypertension and diabetes, and, in contrast with SCORE populations, patients with diabetes were also involved in the Framingham study and the factor of treated hypertension was also registered. Additionally, FRS predicts non-lethal CV events besides CV mortality [2,3]. The differences between the SCORE- and FRS-derived vascular ages in hypertensive and diabetic patients could be explained by these facts. In the case of ePWV, in contrast with risk score-based methods, its calculation is more dependent on the measured blood pressure, so it is more measurement-based, similar to PWV and CACS.

It is also worth mentioning that ePWV is only an estimation, which means that it cannot totally substitute the measurement. Moreover, there are different methods for ePWV calculation as well, with their own limitations. In the Mobil-O-Graph device, the ARCSolver method is used for PWV estimation [19], which is a predictor of CV outcome in patients with suspected coronary artery disease (in line with the ePWV method applied in our present study) [20], but it did not work well in patients with Marfan syndrome [21]. Additionally, in the MORGAM Project which included 107,599 subjects in 38 cohorts from 11 countries, ePWV (the same method like in our present paper) was only associated with all-cause mortality and not CV mortality after adjustment for traditional CV risk factors [22]. These results suggest that in the case of ePWV, similar to vascular age calculation methods, more work is needed to describe its strengths and limitations before its routine use in clinical practice.

This study has limitations that should be considered. This was a cross-sectional study which does not allow us to draw conclusions about the outcome of patients with different vascular age results. Prospective data and head-to-head comparison of different preventive strategies based on different vascular age calculation methods are necessary to clarify which method is the most effective. Additionally, when we determined the vascular age categories, we chose an arbitrary threshold of 2 years’ difference, which is not based on consensus but was previously used in one of our studies [6]. 

## 5. Conclusions

We confirmed our previous results [6], in a population-based cohort, that different vascular age calculation methods can provide different vascular age results and identify different subjects with early vascular ageing. The importance of this finding for implementation in CV preventive strategies requires further studies.

## Figures and Tables

**Figure 1 jcm-13-00205-f001:**
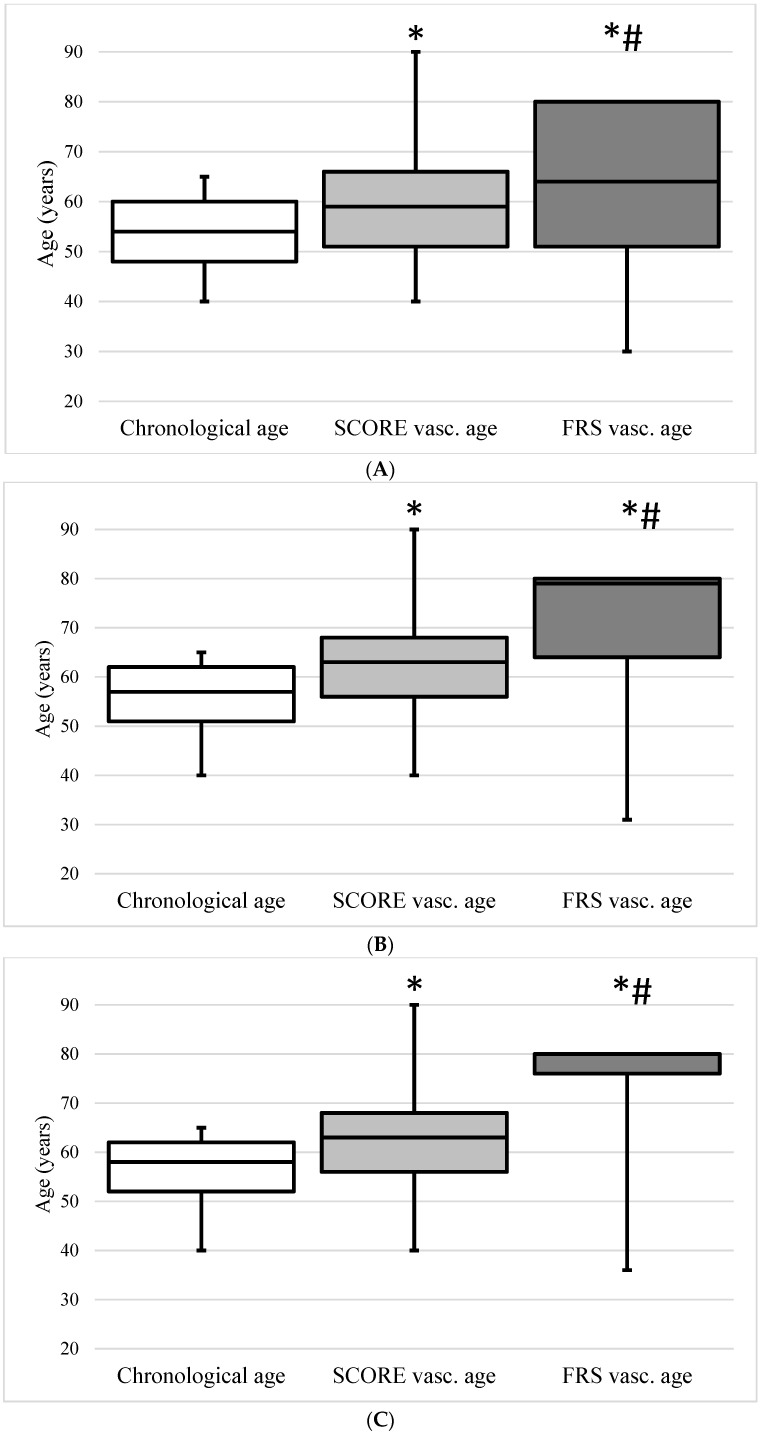
The chronological age and the vascular age calculated based on the Systematic Coronary Risk Evaluation (SCORE vasc. age) and the vascular age calculated based on the Framingham Risk Score (FRS vasc. age) in the total population (**A**), patients with hypertension (**B**), patients with diabetes (**C**), and patients without diabetes or hypertension (**D**). Data are presented as the median (interquartile ranges in error bars). * *p* < 0.05 compared with chronological age; # *p* < 0.05 compared with SCORE vasc. Age.

**Figure 2 jcm-13-00205-f002:**
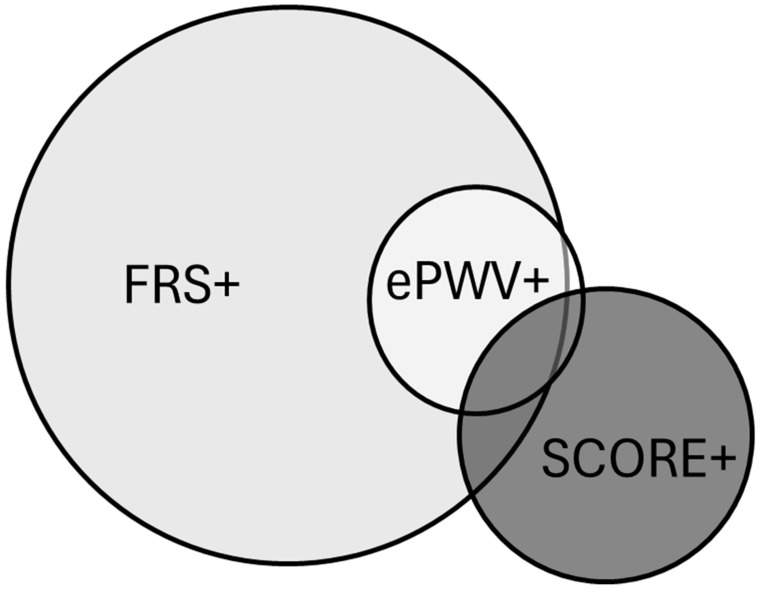
Overlap between subjects identified with early vascular ageing as having more than 2 years of difference compared to their chronological age with the Framingham Score-based method (FRS+), the SCORE-based method (SCORE+), or above 90% with estimated pulse wave velocity (ePWV+).

**Table 1 jcm-13-00205-t001:** Demographic data and baseline laboratory parameters. Data are presented as the median (interquartile ranges). BP, blood pressure, ePWV, estimated pulse wave velocity, and HDL-cholesterol, high-density lipoprotein cholesterol.

n	99,231
Age (years)	54.0 (48.0–60.0)
Men (%)	40,443 (40.8)
Women (%)	58,788 (59.2)
Hypertension (%)	49,191 (49.6)
Diabetes (%)	15,921 (16.0)
Smoking (%)	28,956 (29.2)
Systolic BP (mmHg)	130.0 (122.0–130.0)
Diastolic BP (mmHg)	80.0 (76.0–86.0)
ePWV (m/s)	9.0 (8.1–10.0)
Cholesterol (mmol/L)	5.4 (4.7–6.2)
HDL-cholesterol (mmol/L)	1.4 (1.2–1.7)

**Table 2 jcm-13-00205-t002:** Characteristics of the study participants in relation to their ePWV-based vascular ageing status. Data are presented as the median (interquartile ranges) SCORE vascular age and vascular age based on the Systematic Coronary Risk Evaluation method and FRS vascular ag based on the Framingham Risk Score method. ePWV, estimated pulse wave velocity, and HDL-cholesterol, high-density lipoprotein cholesterol.

	Supernormal Vascular Aging	Normal Vascular Aging	Early Vascular Aging
N (%)	10,557 (10.6)	78,855 (79.5)	9819 (9.9)
Men (%)	2671 (25.3)	32,703 (41.5)	5069 (51.6)
Women (%)	7886 (74.7)	46,152 (58.5)	4750 (48.4)
Chronological age (years)	51.0 (45.0–60.0)	54.0 (48.0–61.0)	57.0 (48.0–62.0)
SCORE vascular age (years)	52.0 (46.0–60.0)	59.0 (51.0–66.0)	67.00 (58.0–74.0)
FRS vascular age (years)	59.0 (48.0–80.0)	64.0 (51.0–80.0)	80.0 (68.0–80.0)
ePWV (m/s)	7.80 (6.9–8.7)	9.0 (8.2–9.9)	10.6 (9.80–11.5)
Hypertension (%)	2892 (27.4)	39,436 (50.0)	6863 (69.9)
Diabetes (%)	935 (8.9)	12,936 (16.4)	2050 (20.9)
No hypertension or diabetes (%)	7375 (69.8)	36,757 (46.6)	2718 (27.7)
Smoking (%)	2844 (26.9)	22,851 (29.0)	3261 (33.2)
Systolic BP (mmHg)	115.0 (110.0–120.0)	130.0 (125.0–140.0)	157.0 (150.0–167.0)
Diastolic BP (mmHg)	70.0 (67.0–72.0)	80.0 (78.0–85.0)	95.00 (90.0–100.0)
Cholesterol (mmol/L)	5.3 (4.6–6.1)	5.4 (4.7–6.2)	5.60 (4.81–6.40)
HDL-cholesterol (mmol/L)	1.5 (1.2–1.8)	1.40 (1.2–1.7)	1.4 (1.2–1.7)

**Table 3 jcm-13-00205-t003:** Overlap between participants identified as having supernormal, normal, or early vascular ageing with the different methods. Data are presented as the median (interquartile ranges); categorical parameters are presented as n (%). SCORE- and FRS- were defined as >2 years younger compared with chronological age. Those between −2–+2 years were defined as normal and those >2 years older were defined as SCORE+ and FRS+.

	Supernormal Vascular Aging	Normal Vascular Aging	Early Vascular Aging
FRS− (n %)	5521 (52.3)	13,077 (16.6)	61 (0.1)
FRS normal (n %)	1554 (14.7)	8854 (11.2)	200 (0.6)
FRS+ (n %)	3482 (33)	56,924 (72.2)	9558 (97.3)
SCORE− (n %)	2776 (26.3)	47,742 (60.5)	6915 (70.4)
SCORE normal (n %)	1868 (17.7)	13,742 (17.4)	1600 (16.3)
SCORE+ (n %)	5913 (56.0)	17,371 (22.1)	1304 (13.3)

## Data Availability

The data presented in this study are available on request from the corresponding author. The data are not publicly available due to privacy.

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
