# Peer review of "Differences between SCORE, Framingham Risk Score, and Estimated Pulse Wave Velocity-Based Vascular Age Calculation Methods Based on Data from the Three Generations Health Program in Hungary"

_jcm, 2023, doi:10.3390/jcm13010205_

Round 1
Reviewer 1 Report
Comments and Suggestions for Authors
The study "Differences between SCORE, Framingham Risk Score and estimated pulse wave velocity-based vascular age calculation methods based on data from the Three Generations Health Program in Hungary" aimed to compare SCORE and FRS-based vascular age calculation methods and their relation to early vascular ageing based on ePWV. Early vascular ageing was defined as ePWV values above the 90th percentile. This cross-sectional study is based on data from almost 100.000 subjects participating in a national primary health care screening program in Hungary. The mean age was 54 years; about half of the participants had arterial hypertension, one-third had a smoking history, and one-sixth had diabetes mellitus.
In brief, the authors report the following in the overall population, as well as in the cohort of patients with arterial hypertension and the cohort of patients with diabetes mellitus:
Chronological age < SCORE-based vascular age < SCORE-based vascular age.
Based on ePWV, FRS identified patients with elevated vascular age with high sensitivity (97%), while SCORE's sensitivity was only 13%. The authors concluded that different vascular age calculation methods can provide different vascular age results in a population-based cohort.
The question is interesting, as "vascular age "has been recommended as a communication strategy in the literature and has found its way into medical guidelines. Therefore, this finding is important. However, I have some comments on the presentation of the issue:
- The period of data collection for this cross-sectional study is not specified. Please add.
- Are outcome data already available or to be expected? A (future?) longitudinal study would be even more meaningful. Please comment on the manuscript.
- Line 251–257: the authors describe differences between the SCORE and FRS-derived vascular ages in hypertensive and diabetic patients. Importantly, when interpreting the results of the patients' cohort with diabetes mellitus, the following must be taken into account:
- FRS: Separate Framingham risk function-based risk charts for diabetic people have been given.
- SCORE: Diabetes mellitus has not been included in the score because data on diabetes had not been collected uniformly in SCORE study cohorts.
4. Score-based, functional and morphological estimates of vascular age are entirely different methods. The comparison is like comparing apples and oranges. In addition, there are different reference cohorts on which the data for determining vascular ages are based. So far, only very few publications have pointed out this fundamental limitation. I refer to the following publications as examples: Yurdadogan, T., et al., Functional versus morphological assessment of vascular age in patients with coronary heart disease. Sci Rep, 2021. 11(1): p. 18164. Sigl, M., et al., Comparison of Functional and Morphological Estimates of Vascular Age. In Vivo, 2023. 37(5): p. 2178-2187. These aspects should be added to the discussion. The publication will gain in importance as a result. As a consequence, the question was even asked whether the use of the "vascular age "is helpful or rather harmful and how credible the information is for the patients (Bonner, C., et al., Is the "Heart Age" Concept Helpful or Harmful Compared to Absolute Cardiovascular Disease Risk? An Experimental Study. Med Decis Making, 2015. 35(8): p. 967-78.).
Author Response
We would like to thank Reviewer 1 for the revision of our manuscript. We were pleased to read that the Reviewer found our results to be important. Our responses for the comments are the following:
- The period of data collection for this cross-sectional study is not specified. Please add.
- 1. We are sorry for this missing data. Data collection was between January 2019 and January 2022. Now it is mentioned at the beginning of the Results session.
- Are outcome data already available or to be expected? A (future?) longitudinal study would be even more meaningful. Please comment on the manuscript.
ad 2. Unfortunately in this screening program no resources were allocated for follow-up data collection. We mention this fact in the “limitations” as “This was a cross- sectional study which does not allow us to draw conclusions about the outcome of patients with different vascular age results. Prospective data and head-to-head comparison of different preventive strategies based on different vascular age calculation methods would be necessary to clarify which method is the most effective.”
- Line 251–257: the authors describe differences between the SCORE and FRS-derived vascular ages in hypertensive and diabetic patients. Importantly, when interpreting the results of the patients' cohort with diabetes mellitus, the following must be taken into account:
- FRS: Separate Framingham risk function-based risk charts for diabetic people have been given.
- SCORE: Diabetes mellitus has not been included in the score because data on diabetes had not been collected uniformly in SCORE study cohorts.
ad 3. We would like to thank the Reviewer this comment. We modified the sentences of the discussion as the following: “Otherwise, FRS integrates treated risk factors as well, like hypertension and diabetes as, in contrast with SCORE populations, patients with diabetes were also involved into Framingham study and the fact of treated hypertension was also registered. Additionally, FRS predicts the non-lethal CV events besides CV mortality (2) (3).”
4 Score-based, functional and morphological estimates of vascular age are entirely different methods. The comparison is like comparing apples and oranges. In addition, there are different reference cohorts on which the data for determining vascular ages are based. So far, only very few publications have pointed out this fundamental limitation. I refer to the following publications as examples: Yurdadogan, T., et al., Functional versus morphological assessment of vascular age in patients with coronary heart disease. Sci Rep, 2021. 11(1): p. 18164. Sigl, M., et al., Comparison of Functional and Morphological Estimates of Vascular Age. In Vivo, 2023. 37(5): p. 2178-2187. These aspects should be added to the discussion. The publication will gain in importance as a result. As a consequence, the question was even asked whether the use of the "vascular age "is helpful or rather harmful and how credible the information is for the patients (Bonner, C., et al., Is the "Heart Age" Concept Helpful or Harmful Compared to Absolute Cardiovascular Disease Risk? An Experimental Study. Med Decis Making, 2015. 35(8): p. 967-78.).
ad 4. We would also like to thank this comment to Reviewer 2. We suppose, besides measurement-based functional and morphological vascular age calculation methods, risk score-based ones represent a third category. We added the following sentences to the Discussion session: “Vascular age calculation methods can also be divided into functional and morphological measurement-based categories, besides risk score-based ones. In line with our findings, recent publications discovered marked differences between functional and morphological methods as well. Both in the study of Yurdadogan T et al. and Sigl M et al., in high proportion of subjects, significant differences were found between the PWV-based (functional) and carotid artery intima-media thickness-based (morphological) methods in the identification of early vascular ageing (refs). Besides these emerging problems, the hypothesis, as vascular ageing communication is superior than communication of the absolute CV risk in percentage is not proven yet. Moreover, in low-risk patients, and internet-based survey with a short-term follow-up (2 weeks) failed to prove the superior effectiveness of vascular age communication for intention to change lifestyle (ref).”
Reviewer 2 Report
Comments and Suggestions for Authors
This is an interesting research which aims to offer information regarding different scores used for stratifying the cardiovascular risk and their relation with the estimated vascular age. The particular aspect of this topic is that different scores identify different populations which are to a greater or lesser extent related to their vascular age according to estimated pulse wave velocity. The results are valuable in the light of the primary prevention.
This is generally a well-written article. I would only suggest to summarize the results presented at lines 211-217 into a Venn diagram or another suitable graphic form to emphasize the relation between the used scores and the vascular age.
The authors correctly identified the limitation of this cross-sectional study.
Author Response
We would like to thank Reviewer 2 for the revision of our manuscript. As requested, we made a new figure (Figure 2) to demonstrate the overlap between the different vascular age calculation methods in a Venn diagram. We think, the paper improved with this suggestion. We hope, that after this modification of the manuscript it will be acceptable for the Reviewer.